# A Small Molecule Promoting Neural Differentiation Suppresses Cancer Stem Cells in Colorectal Cancer

**DOI:** 10.3390/biomedicines10040859

**Published:** 2022-04-06

**Authors:** Jung Kyu Choi, Ihn-Sil Kwak, Sae-Bom Yoon, Heeyeong Cho, Byoung-San Moon

**Affiliations:** 1Department of Biotechnology, College of Life and Applied Sciences, Yeungnam University, Gyeongsan 38541, Korea; jungkyuc@ynu.ac.kr; 2Department of Ocean Integrated Science, Chonnam National University, Yeosu 59626, Korea; iskwak@jnu.ac.kr; 3Therapeutics and Biotechnology Division, Drug Discovery Platform Research Center, Korea Research Institute of Chemical Technology (KRICT), Daejeon 34114, Korea; bomi9123@krict.re.kr (S.-B.Y.); hycho@krict.re.kr (H.C.); 4Department of Biotechnology, Chonnam National University, Yeosu 59626, Korea

**Keywords:** colorectal cancer, cancer stem cell, neural progenitor cell, Wnt/β-catenin, *K-Ras*

## Abstract

Cancer stem cells (CSCs) are a tumor cell subpopulation that drives tumor progression and metastasis, leading to a poor overall survival of patients. In colorectal cancer (CRC), the hyper-activation of Wnt/β-catenin signaling by a mutation of both adenomatous polyposis coli (APC) and *K-Ras* increases the size of the CSC population. We previously showed that CPD0857 inactivates Wnt/β-catenin signaling by promoting the ubiquitin-dependent proteasomal degradation of β-catenin and Ras proteins, thereby decreasing proliferation and increasing the apoptosis of CRC lines. CPD0857 also decreased the growth and invasiveness of CRC cells harboring mutant *K-Ras* resistant to EGFR mAb therapy. Here, we show that CPD0857 treatment decreases proliferation and increases the neuronal differentiation of neural progenitor cells (NPCs). CDP0857 effectively reduced the expression of CSC markers and suppressed self-renewal capacity. CPD0857 treatment also inhibited the proliferation and expression of CSC markers in D-*K-Ras* MT cells carrying *K-Ras*, *APC* and *PI3K* mutations, indicating the inhibition of PI3K/AKT signaling. Moreover, CPD0857-treated xenograft mice showed a regression of tumor growth and decreased numbers of CSCs in tumors. We conclude that CPD0857 could serve as the basis of a drug development strategy targeting CSCs activated through Wnt/β-catenin-Ras MAPK-PI3K/AKT signaling in CRCs.

## 1. Introduction

Colorectal cancer (CRC) is one of the most common cancers and the second-leading cause of cancer-related deaths worldwide [1]. CRC is a stem cell disease that occurs when an intestinal stem cell (ISC) escapes control and gives rise to a cancer stem cell (CSC) [2,3,4,5,6]. CSCs, also known as tumor-initiating or tumor-propagating cells, are a tumor cell subpopulation that can self-renew and differentiate into new tumors; thus, they contribute to tumorigenesis, metastasis and relapse [1,7,8,9,10,11]. Consequently, identifying and targeting factors involved in CSC activation has emerged as a promising CRC therapy [5,12].

The intestinal crypt stem cell niche, a unique microenvironment that enables optimal stem cell function, contains numerous molecular and cellular effectors [13,14]. If aberrant signaling is activated and maintained in that niche, neoplasias may emerge [13,15]. CRC occurs through a genetic transformation in the Wnt/β-catenin and RAS/ERK signaling pathways, both of which maintain normal ISC self-renewal and proliferation [5,15]. The Wnt/β-catenin pathway in particular supports ISCs and intestinal homeostasis [15,16]. Approximately 90% of CRC patients harbor loss-of-function mutations in the gene that encodes adenomatous polyposis coli (APC) [17,18], a mutation that promotes the nuclear accumulation of the Wnt effector β-catenin and leads to the clonal expansion of ISCs and CRC initiation [17,19,20]. Subsequent adenoma progression is induced by a secondary oncogenic *K-Ras* mutation, which occurs in 40–50% of CRC [19,21]. The *K-Ras* mutation does not by itself result in CSC activation [22,23]. However, oncogenic *K-Ras* mutation results in CSC activation involving malignant conversion in the presence of an *APC* mutation [24,25]. The regulation of oncogenic *K-Ras* through the Wnt/β-catenin pathway is key for the interaction between the Wnt/β-catenin and RAS/ERK signaling pathways [5,25]. 

Neural cells play an important role in tumor growth, invasion and metastasis and are considered components of the cancer microenvironment [1,26,27]. Cancer cells attract nerve fibers and stimulate nerve outgrowth by secreting neurotrophic factors [1,27,28,29]. Conversely, nerve fibers can infiltrate the tumor microenvironment and promote tumor growth and cancer cell dissemination [1,30]. Additionally, various pleiotropic signaling molecules control the self-renewal and differentiation of neural progenitor cells (NPCs) [1,31]. Thus, targeting cancer and neurogenesis may be promising in terms of the development of cancer therapy. Therefore, we asked whether a compound that we previously identified could induce the neuronal differentiation of NPCs to suppress tumorigenesis and inhibit the growth of CRC-mediated tumor microenvironments. 

Previously, we performed a high-throughput screening (HTS) using a dual-cell-based screening system to identify a small molecule that decreased the levels of both β-catenin and Ras proteins by inhibiting Wnt/β-catenin and Ras/ERK signaling [32]. Using this system, we identified CPD0857 as an active compound. Thus, we asked whether CPD0857 could promote the neuronal differentiation of NPCs and inhibit tumor growth and progression in vivo and in vitro. Here, we show that CPD0857 effectively promotes the neuronal differentiation of NPCs compared to other chemotherapeutic agents and significantly reduces the subpopulation of CRC causing cancer drug resistance through the inhibition of Wnt/β-catenin-Ras MAPK-PI3K/AKT signaling.

## 2. Materials and Methods

### 2.1. Cell Culture

CRC cell lines, including HT29, HCT116, HCT15, SW480, Caco2, RKO and LoVo, were purchased from the American Type Culture Collection (ATCC, Manassas, VA, USA). Isogenic cell lines (DLD-1 (D)-*K-Ras* WT, D-*K-Ras* MT & *PI3K* WT, and D-*K-Ras* MT & *PI3K* MT) were provided by B. Vogelstein (John Hopkins University School of Medicine, Baltimore, MD, USA). HEK293 and HEK293-TOP flash reporter cells were grown in DMEM (Gibco) containing 10% fetal bovine serum (FBS), 100 U/mL penicillin, and 10 μg/mL streptomycin at 37 °C. Human CRC lines were grown in RPMI 1640 (Gibco) containing 10% fetal bovine serum (FBS), 100 U/mL penicillin, and 10 μg/mL streptomycin at 37 °C. 

### 2.2. Sphere Forming Assay

Serum-free medium used for sphere culture was composed of DMEM/F12 medium supplemented with 100 IU/ml penicillin, 10 μg/ml streptomycin, and 20 ng/ml human recombinant epidermal growth factor, 20 ng/ml human recombinant basic fibroblast growth factor, and 2% B27 supplement (Invitrogen, USA). CRC cells were seeded at a density of 2 × 10^4^ (*n* = 3) cells/plate in ultra-low attachment 6-well plates for 5 days, and then treated with varying drug doses for 72 h. Relative numbers and sizes of colospheres were measured using ImageJ software. 

### 2.3. Screen for Drugs That Promote Neuronal Differentiation

Neural progenitor cells (NPCs) were surgically extracted from the forebrain of E14.5 rats and maintained in an undifferentiated state by culture in DMEM/F12 with 20 ng/mL bFGF (Peprotech). Among the phenotypes analyzed were cell number and morphology, as assessed by analysis of phase contrast images of cells after 48 h. 

### 2.4. High-Contents Screening 

The Cellomics Multiparameter Neurite-Outgrowth Kit was used to perform multiple neuronal differentiation assays. According to their protocols, NPCs were plated in a 96-well plate at a concentration of 2 × 10^4^ cells per well, grown in medium in the presence or absence of bFGF (20 ng/ml), and then treated with AG1478 (EGFR tyrosin kinase inhibitor, 10, 20 μM), Cetuximab (EGFR monoclonal antibody, 5, 10 μM), 5-FU (antineoplastic drug, 10, 20 ug/mL), LY294002 (PI3K/AKT inhibitor, 10, 20 μM), and CDP0857 (10, 25 μM) during 24 h. Next, 50 μL of paraformaldehyde 4% was added to each well to fix the cells, and PBS was used to wash the cells. Then, the cells were incubated with 30 μL of primary antibody solution per well for 2 h. The cells were further incubated for 1 h with 30 μL of secondary antibody solution at room temperature. Nuclei were counterstained with DAPI. Cellular screening, data acquisition, and data interpretation were conducted on the ArrayScan HCS Reader (Cellomics Inc., Pittsburgh, PA, USA).

### 2.5. Immunoblotting Assay

Cells or tissues were gently lysed in RadioImmunoPrecipitation Assay (RIPA) buffer (Upstate Biotechnology, Lake Placid, NY, USA) for 1 h on ice and centrifuged at 12,000 rpm at 4 °C for 15 min. Lysates were boiled for 5 min at 95 °C in SDS sample buffer and separated on 10% SDSPAGE. After blocking, membranes were incubated first with primary antibodies and then with a peroxidase-conjugated secondary antibody. Bound secondary antibody (anti-mouse or anti-rabbit 1:10,000) was detected using the enhanced chemiluminescence (ECL) reagent (Santa Cruz Biotechnology). Images were taken using the luminescent image analyzer LAS-3000 (Fujifilm, Tokyo, Japan).

### 2.6. Immunohistochemistry and Immunocytochemistry

For immunohistochemistry, xenograft tissues were dissected and fixed in 4% paraformaldehyde (PFA) at 4 °C. Paraffin sections were incubated with primary antibody at 4 °C for 18 h. For immunocytochemistry, cells cultured on coverslips were fixed with 4% PFA/PBS for 2 h and immune-stained after permeabilizing with 0.2% Triton X-100. Cells and tissues were then incubated with indicated primary antibodies overnight at 4 °C, followed by Alexa Fluor 488 (Life Technologies, Carlsbad, CA, USA) or Alexa Fluor 555 (Life Technologies) secondary antibodies at room temperature for 1 h and counterstained with 4′-6′diamidino-2-phenylindole (DAPI) (Boehringer Mannheim, Mannheim, Germany). Images were visualized using confocal microscopy (LSM5 PASCAL; Zeiss, Jena, Germany). Values obtained from at least three independent experiments were averaged and reported as mean ± SD. Student’s two-tailed t test was used to compare two experimental groups.

### 2.7. Fluorescence-Activated Cell Sorting Analysis

For FACS, cells were gently dissociated and filtered through cell strainers (70 μm; BD Falcon) followed by incubation with conjugated primary antibodies. Cell surface markers were analyzed with a FACS Vantage SE flow cytometer (BD Biosciences) equipped with FlowJo software (TreeStar), and 10,000 events were recorded for each sample.

### 2.8. Animal Studies

Animal experimental procedures were approved by the Institutional Animal Care and Use Committee of the Yonsei Laboratory Animal Research Center. BALB/c nu/nu mice were purchased from the Central Lab Animal Inc. (Seoul, Korea). All animals were housed in filter-topped shoebox cages equipped with a computerized environmental control system. Room temperature was maintained at 24 °C with 40–70% relative humidity. After acclimatization for 1 week, mice were subcutaneously injected with 1 × 10^6^ DLD-1-*K-Ras* mutant (D-*K-Ras* MT) cells in 200 μL phosphate-buffered saline/Matrigel (1:1) in the dorsal flank. When mean tumor volumes reached ~ 200 mm^3^, mice were randomly divided into two groups (four per group) and administered either CPD0857 suspended in 0.5% methyl cellulose/0.5% Tween 80 or vehicle intraperitoneally at a drug dose of 25 mg/kg, twice a week. Tumor volume was measured every 3–4 days using Vernier calipers, applying the formula: π/6 × length × width × height. Animals were euthanized when tumor volume exceeded 1500 mm^3^. Tumors were then excised, weighed, and fixed in 4% paraformaldehyde for further analysis.

### 2.9. Antibodies and Reagents

Antibodies used in this study were anti-β-actin (Santa Cruz Biotechnology, Santa Cruz, CA, USA), anti-PCNA (Santa Cruz Biotechnology), anti-TUJ1 (Abcam Inc., Cambridge, MA, USA), anti-Nestin (Abcam Inc., Cambridge, MA, USA), anti-Ki67 (Sigma-Aldrich, St. Louis, MI, USA), anti-BrdU (Sigma-Aldrich, St. Louis, MI, USA), anti-CD133 (Sigma-Aldrich, St. Louis, MI, USA), anti-CD44 (Abcam Inc., Cambridge, MA, USA), anti-EpCAM (Abcam Inc., Cambridge, MA, USA), anti-panRas (Upstate Biotechnology, Lake Placid, NY, USA; Abcam Inc., Cambridge, MA, USA), anti-β-catenin (Cell Signaling Biotechnology, Beverly, MA, USA), anti-pERK (Thr202/Tyr204) (Cell Signaling Biotechnology, Beverly, MA, USA), anti-BrdU (Sigma-Aldrich, St. Louis, MI, USA), and HRP-conjugated anti-mouse (Bio-Rad Laboratories, Hercules, CA, USA). Fluorescent anti-rabbit secondary antibodies (Calbiochem, La Jolla, CA, USA) were used for detection by a luminescent image analyzer, LAS-3000 (Fujiflm, Tokyo, Japan).

### 2.10. Statistical Analysis

Statistical analyses were performed using Excel statistical tools or Prism 5 (GraphPad Software), where differences between treatment groups were experimentally hypothesized, and statistical differences between two groups were analyzed using Student’s *t*-test (* *p* < 0.05, ** *p* < 0.005, and *** *p* < 0.0005). ANOVA tests (Tukey’s multiple comparison test) were used to test hypotheses about the effects in multiple groups.

## 3. Results

### 3.1. Neuronal Differentiation Activity of Various Compounds following Inhibition of Wnt/β-Catenin Signaling

Colorectal cancer (CRC) is generally characterized by higher β-catenin and Ras expression due to gene mutation [25,33]; thus, identifying an inhibitor of both could eradicate aggressive and metastatic CRC. To screen for such compounds, we used the dual-cell-based screening system previously described [25,33]. Specifically, we used a 2000-compound ChemDiv drug screening library platform to identify compounds that would decrease TOP-flash reporter activity in the HEK293 TCF/LEF cell line. We acquired 100 compounds that inhibited reporter activity and then narrowed them down to the 9 most- and least-effective toxic candidates. We then asked whether any of the nine candidates promoted neuronal differentiation. As shown in Figure 1, to do this, we assessed the neuronal differentiation of primary NPCs derived from the E14.5 rat embryo forebrain in the presence or absence of the basic fibroblast growth factor (bFGF) of the active compound. Overall, compounds CPD966, CPD0004, CPD0857 and CPD2677 promoted NPC differentiation, despite co-treatment with bFGF (Figure 1A). Among the compound candidates, CPD0857 exhibited the most potent differentiation effect based on the analysis of neurite outgrowth (Figure 1B). We then treated NPCs with each of the nine candidates plus valproic acid (VPA), either alone or combined with bFGF. VPA was used for the positive control of neuronal differentiation of NPCs (Figure 1C). CPD0857 treatment reduced proliferating cell nuclear antigen (PCNA) activity and increased levels of the neuron-specific marker Tuj1, as shown by immunoblots (Figure 1C) and band intensity quantification (Figure 1D). To determine whether CPD0857 inhibited TOP-flash reporter activity we co-transfected NPCs with TOP-flash reporter plasmid and β-galactosidase and treated cells with bFGF in the presence or absence of compound. We observed a decreased luciferase reporter activity by close to a half in the presence of CPD0857/bFGF relative to bFGF alone (Figure 1E). Given that the CPD0857 treatment of cancer cells decreased β-catenin and Ras protein expression through ubiquitin-dependent proteasomal degradation in previous studies [33], we focused on β-catenin and Ras protein stability by treating NPCs with CPD0857 dose dependently and assessing β-catenin and Ras protein levels. Treatment with CPD0857 significantly increased phosphor (p)-β-catenin and reduced levels of β-catenin and Ras protein, including pAKT, which is the downstream signaling factor of Ras (Figure 1F,G). Moreover, to confirm these effects, we transiently overexpressed HA-Ubiquitin (HA-Ub) vector in the NPCs and then treated cells with CPD0857 plus MG132 (10 μg/mL). The immunoprecipitation (IP) analysis of endogenous β-catenin and Ras proteins followed by anti-HA immunoblot indicated that both endogenous proteins were polyubiquitylated, and that polyubiquitylated proteins accumulated in MG132-treated NPCs (Figure 1H–J).

### 3.2. CPD0857 Treatment Alters NPC Proliferation and Differentiation

We next determined whether CPD0857 blocked NPC proliferation induced by bFGF. The intermediate filament protein Nestin is an NPC marker expressed in undifferentiated progenitor cells in the developing central nervous system (CNS) [1,34]. As shown in Figure 2A–C, and Appendix A, a high contents screening (HCS) analysis of Nestin and Ki67 staining indicated that CDP0857 reduced proliferation and increased differentiation in NPCs more potently than AG1478, cetuximab, 5-fluoruracil (5-FU) and LY294002 treatment did during one DIV (day in vitro) for all reagents used in various chemotherapies (Figure 2A–C). We also confirmed that combining CDP0857 with bFGF had a greater effect on NPC differentiation than bFGF alone during two DIV based on immunofluorescent analysis of nestin (Figure 2D,E). An RT-PCR analysis also indicated that *nestin* mRNA levels in the presence of CDP0857 with bFGF were significantly reduced compared to treatment with bFGF only (Figure 2F). Moreover, CDP0857 plus bFGF treatment of NPCs decreased BrdU incorporation (indicating decreased proliferation) and increased Tuj1 levels (Figure 2F), and proportions of BrdU-positive cells decreased, while numbers of Tuj1-positive cells increased following co-treatment with CPD0857 plus bFGF (relative to treatment with bFGF only (Figure 2G–I).

### 3.3. Effects of CPD0857 on CRC Cells Carrying K-Ras, APC and PI3K Mutations

Oncogenic *K-Ras* plays a crucial role in CSC activation [25,33]. We observed that both the size and number of spheres (SPs) significantly decreased in a dose dependent manner in SPs of D-*K-Ras* MT cells harboring *APC* and *PI3K* mutations following CPD0857 treatment (Figure 3A,B). After confirming that these were D-*K-Ras* MT cells based on the high expression of CSC markers, CD44 and CD133, we observed a lower expression of CSC markers in SPs treated with CPD0857 relative to vehicle controls (Figure 3C,D). We then used fluorescence-activated cell sorting (FACS) to quantify CD44- or CD133-positive cells in SPs and found that the number of CD44-positive cells in CDP0857-treated SPs was 32.7% less than that found in vehicle controls. Consistently, the number of CD133-positive cells in CDP0857-treated SPs decreased by 34.4% (Figure 3E).

### 3.4. CPD0857 Inhibits Tumorigenesis in CSCs Harboring *K-Ras*, *APC* and *PI3K* Mutations

To investigate CPD0857 activity in colon cancer lines, we examined and quantified CD44 and CD133 expressions in various colon cancer lines, such as CCD-18Co, HCT15, HT-29, (Caco2, SW480, SW480APC15, HCT116 (HAE6) MT, HCT116 HAE6) WT, D-*K-Ras* WT, SW480APC38, HCT116 and D-*K-Ras* MT (Figure 4A). We observed significantly increased staining for both CD44 and CD133 in the D-*K-Ras* MT line (Figure 4A). Therefore, we performed mouse xenograft studies using D-*K-Ras* MT cells harboring *APC* and *PI3K* mutations and treated mice in vivo with CPD0857 (intraperitonially (i.p.), 25 mg/kg). As shown in Figure 4B, CPD0857 administration significantly reduced the size of D-*K-Ras* MT tumors relative to vehicle-treated xenograft mice, and the tumor volume in vehicle-treated mice was much greater than that seen in CPD0857-treated mice (Figure 4C). We then tested CPD0857 effects on Wnt/β-catenin and Ras/ERK signaling using tumors excised from CPD0857- or vehicle-treated xenograft mice. CPD0857 treatment significantly decreased levels of activated proteins found in both pathways, such as β-catenin, Pan-Ras, and p-ERK (Figure 4D). Additionally, the expression of PCNA, CD133, CD44 and EpCAM (epithelial cell adhesion molecule) decreased relative to controls following CPD0857 treatment, based on an immunohistochemical analysis of tumors generated by xenografts (Figure 4E).

## 4. Discussion

*K-Ras* mutations play an important role in promoting the characteristics of CSCs, which are a critical cause of malignancies leading to lethality in CRC patients [1,21,35]. CSC induction by both *APC* and *K-Ras* mutations synergistically activates CRC tumorigenesis [1,21,25]. The Ras/extracellular-signal-regulated kinase (ERK) pathway interacts with Wnt/β-catenin signaling during CRC tumorigenesis and metastasis, and levels of both β-catenin and RAS proteins positively correlate with each other in CRC-harboring *APC* mutations [1,21]. There are several reports that APC and *K-Ras* mutations co-occur at different stages of CRC tumorigenesis and metastasis [25,33]. Moreover, recent studies support the idea that the interaction of Wnt/β-catenin with Ras/ERK pathways during CSC tumorigenesis increases malignancy [25]. Generally, *K-Ras* regulates ERK and phosphatidyl inositol 3-kinase (PI3K)-Akt signaling pathways [25,36]. *K-Ras* mutations alone do not significantly induce the malignant transformation of the mouse intestinal epithelium; however, additional *APC* mutations synergistically enhance tumor growth and metastasis in the liver and lungs [22,25,36,37]. Moreover, patients with metastatic CRC-bearing *K-Ras* mutations overexpress epidermal growth factor receptor (EGFR), a transcriptional target of the Wnt/β-catenin pathway, and are resistant to EGFR mAb drugs, such as cetuximab and panitumumab used as CRC chemotherapy [1,38,39] Thus, it is necessary to develop drugs that simultaneously target the Wnt/β-catenin and Ras/ERK pathways for effective CRC treatment. In this study, we demonstrate that CDP0857 is a chemical compound that inhibits CSCs of CRC tumors by inhibiting Wnt/β-catenin and Ras/ERK pathways [32]. 

Infiltration of the cancer microenvironment by nerve fibers occurs with cancer progression and is a sign of poor prognosis [27]. Moreover, the CRC microenvironment is rich in autonomic nerve fibers [25,36,40]. Recent studies showed that a fraction of CSCs derived from patients with colorectal carcinoma can generate neurons that function in tumor neurogenesis and tumor growth [27]. To assess whether CDP0857 regulates NPC proliferation, we used primary NPCs derived from the E14.5 rat embryo forebrain. The CDP0857 treatment of these cells effectively induced neuronal differentiation. Basic fibroblast growth factor (bFGF) modulates neurogenesis and NPC proliferation, in vivo and in vitro [25,36,41]. The treatment of NPCs with CDP0857 resulted in a phenotype similar to that seen in the absence of bFGF, namely, a differentiation marked by neurite outgrowth and decreased proliferation. Valproic acid (VPA) reportedly induces NPC differentiation and inhibits growth [25,36,42]; CDP0857 treatment reduced PCNA activity and BrdU incorporation and increased the expression of the neuronal marker Tuj1 levels beyond the effects seen in the presence of VPA. Additionally, based on the staining of the NPC marker nestin, CDP0857 decreased the proliferation and increased the differentiation of NPCs more potently than the commonly used anticancer drugs AG1478, cetuximab, 5-fluoruracil (5-FU) and LY294002. These results indicate that CDP0857, which inhibits Wnt/β-catenin signaling, suppresses NPC proliferation and promotes neuronal differentiation. 

CDP0857 also inhibited the self-renewal capacity of CRC cells in SPs of D-*K-Ras* MT cells harboring *APC* and *PI3K* mutations. Likewise, CDP0857 treatment of various colon cancer lines decreased levels of the cell-surface proteins CD44 and CD133, which are CSC markers. To confirm these results in vivo, we performed mouse xenograft studies using D-*K-Ras* MT cells harboring *APC* and *PI3K* mutations and treated mice with CDP0857. CDP0857 treatment significantly reduced the tumor size and volume of D-*K-Ras* MT cells harboring these mutations, and tumors excised from xenograft mice showed a significantly reduced expression of CD44, CD133 and EpCAM and of active forms of factors in the Wnt/β-catenin and Ras/ERK pathways. Additionally, CDP0857 treatment decreased the expression of proliferation markers. Overall, although additional research is needed to define the relative contributions of the pathways involved, our results suggest that CDP0857 could serve as a potential therapeutic agent to inhibit CSC activation in CRC associated with neuronal differentiation.

## Figures and Tables

**Figure 1 biomedicines-10-00859-f001:**
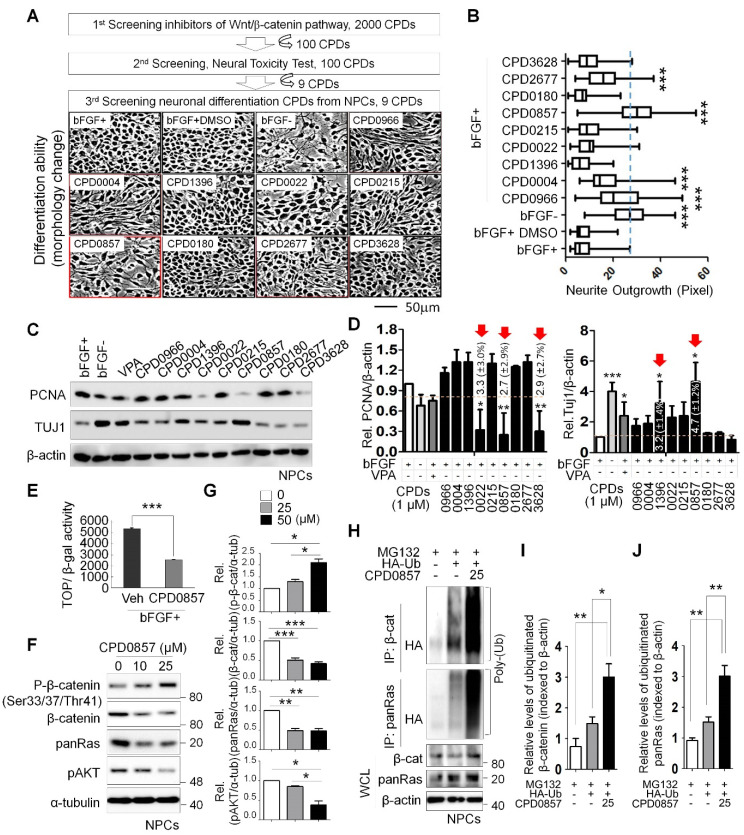
Screening drugs promoting neuronal differentiation of the NPCs via inhibiting Wnt/β-catenin signaling. (**A**) Nine lead compounds selected by a dual high-throughput screening system, which was described previously [32], were tested for measuring neuronal differentiation using primary NPCs derived from the E14.5 rat embryo forebrain. NPCs were treated with 9 compounds (10 μM) for 48 h in the presence or absence of 20 ng/ml bFGF. Micrographs were taken at a magnification of 200×. Scale bars: 50 μm. (**B**) Neurite outgrowths were quantified using ImageJ software. All data are shown as mean ± SD for at least 3 independent images. *** *p*  < 0.005, 2-sided t test. (**C**) NPCs were grown for 18 h in N2:B27 (1:1) medium containing bFGF and treated with VPA (negative control, 1mM) and each compound (10 μM). WCLs were immunoblotted using indicated antibodies. (**D**) Relative band intensity shown in (**C**) were quantified using ImageJ software (*n* = 3). Red arrows indicate significant decreases or increases in PCNA or Tuj1 in NPCs, respectively. (**E**) NPCs were transfected with Topflash reporter plasmid and co-transfected with β-galactosidase. Then, 24 h later, the growth medium containing transfection reagents was removed and replaced with N2 media with or without 25 μM CDP0857 for a further 24 h. Cells were then harvested and luciferase reporter assays were performed. (**F**) Western blot (WB) analysis of NPCs using indicated antibodies. (**G**) Relative band intensities shown in (**F**) were quantified using ImageJ software (*n* > 2). (**H**) Ubiquitylation assay of β-catenin and Ras proteins in indicated lysates of NPCs. Cells were transduced with HA-Ub plasmids and then treated a day later with or without CPD0857 for 24 h. Cells were also treated with MG132 (10 μg/mL) during 6 h before harvest. Immunoprecipitation (IP) was performed with β-catenin or Ras antibodies. WCLs were analyzed by WB for indicated antibodies (*n* = 2). (**I**,**J**) Relative band intensity of ubiquitylated proteins shown in (**H**) were quantified using ImageJ software (*n* = 3). Values correspond to mean ± SD. ANOVA tests were performed to calculate significance (* *p* < 0.05, ** *p* < 0.005, *** *p* < 0.005). NPCs, neural progenitor cells; VPA, valproic acid; WCLs, whole cell lysates; Ub, ubiquitin.

**Figure 2 biomedicines-10-00859-f002:**
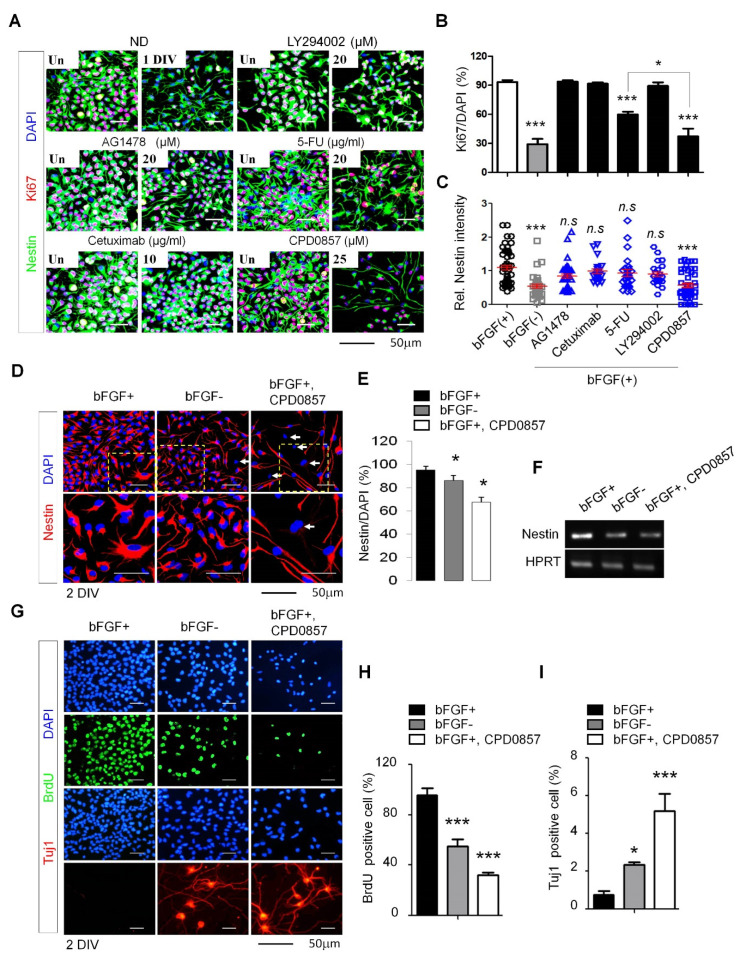
The effect of CDP0857 on proliferation and differentiation of the NPCs. (**A**) NPCs were grown in medium in the presence or absence of bFGF (20 ng/mL) and were treated with AG1478 (EGFR tyrosin kinase inhibitor, 20 μM), Cetuximab (EGFR monoclonal antibody, 10 μM), 5-FU (antineoplastic drug, 20 ug/mL), LY294002 (PI3K/AKT inhibitor, 20 μM), and CDP0857 (25 μM) during 24 h. Cells were subjected to immunofluorescent labeling using antibody specific for Nestin. Nuclei were counterstained with DAPI. Images were obtained using a Zeiss confocal microscope. Scale bars: 50 μm. (**B**) Quantification of Ki67/DAPI cells in analysis shown in (**A**, right) (*n* = 3). (**C**) The intensity of Nestin (blue) was quantified by using ImageJ software (*n* > 3). (**D**) NPCs were treated with CDP0857 (25 μM) during 48 h. Immunostaining with Nestin (red) antibody. DAPI (blue). Images were obtained using a Zeiss confocal microscope. (**E**) Quantification of Nestin/DAPI cells in analysis shown at (**D**) (*n* = 3). (**F**) RT-PCR analysis of indicated mRNAs in analysis shown in (**D**). (**G**) NPCs were treated with CDP0857 (25 μM) during 48 h. Proliferation and neuronal differentiation was monitored by BrdU incorporation and Tuj1 staining using immunocytochemical analysis. Nuclei were counterstained with DAPI. Scale bars: 50 μm. (**H**,**I**) Percentages of relative numbers of BrdU-positive cells (**H**) and Tuj1-positive cells (**I**). Values correspond to mean ± SD. ANOVA tests were performed to calculate significance (* *p* < 0.05, *** *p* < 0.0005). bFGF, basic fibroblast growth factor; BrdU, 5-bromo-2-deoxyuridine; DAPI, 40,6-diamidino-2-phenylindole.

**Figure 3 biomedicines-10-00859-f003:**
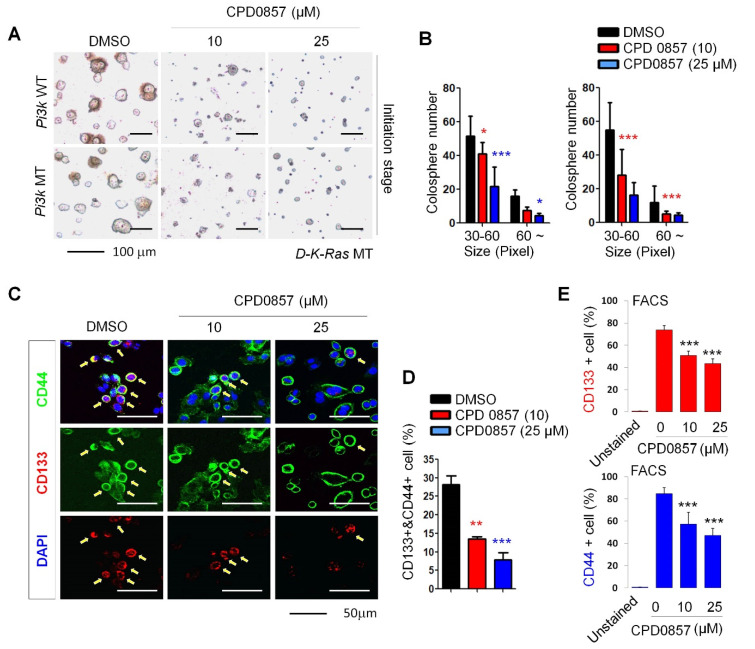
Effects of CDP0857 on sphere formation and cancer stem cell activation of colorectal cancer (CRC) cells carrying *K-Ras*, *APC* and *PI3K* mutations. (**A**–**C**) D-*K-Ras* WT and D-*K-Ras* MT cells harboring mutations of both *APC* and *PI3K* were seeded at a density of 2 × 10^4^ (*n* = 3) cells/plate in ultra-low-attachment 6-well plates for 5 days, and were then treated with varying doses of CDP0857 for 72 h. Scale bars, 100 μm. (**B**) Relative numbers and sizes of colospheres were measured using ImageJ software. (**C**) D-*K-Ras* MT cells harboring mutation both *APC* and *PI3K* were seeded at a density of 2 × 10^4^ cells/well in 24-well plates and were treated with varying doses of CDP0857 for 24 h. Immunofluorescent labeling of CD133 and CD44. Nuclei were counterstained with DAPI. Images were obtained using a Zeiss confocal microscope. Scale bars, 50 μm. (**D**) Quantification of CD133 and CD44-double-positive populations in each immunostained tumor cell. (**E**) The relative numbers of CD133- or CD44-positive cells were quantified by fluorescent-activated cell sorter (FACS) analyses. WT, wild-type; MT, mutant-type. * *p* < 0.05, ** *p* < 0.005, *** *p* < 0.005.

**Figure 4 biomedicines-10-00859-f004:**
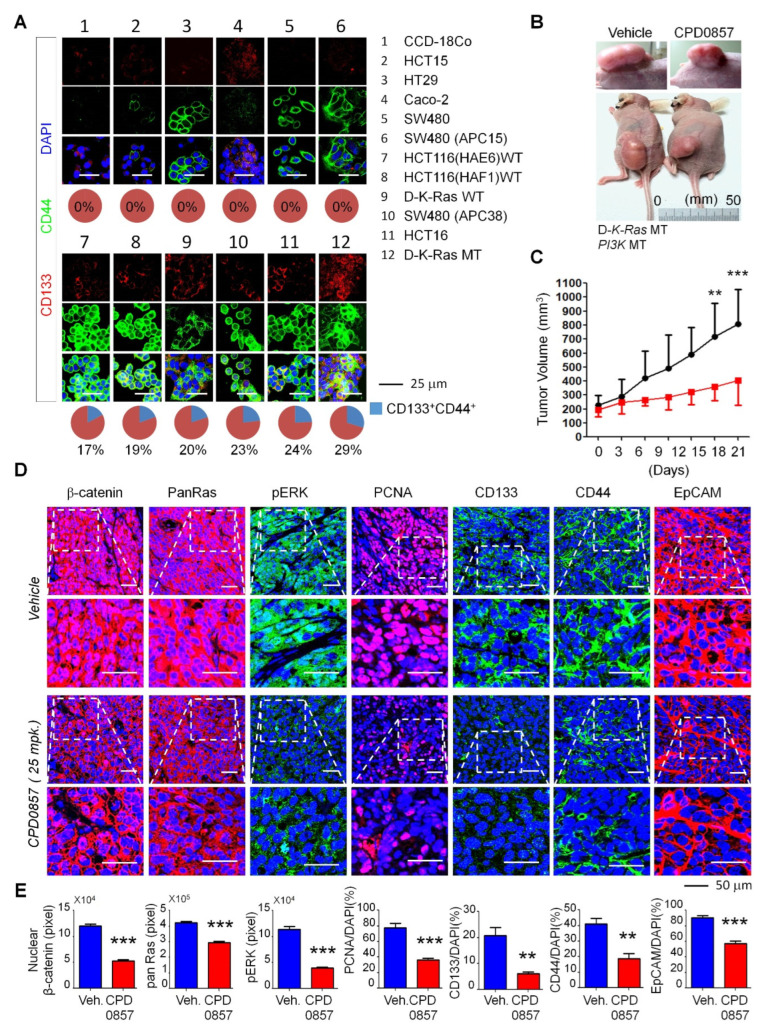
CDP0857 inhibits progression of tumor with mutations of *K-Ras*, *APC* and *PI3K* by reducing cancer stem cell populations. (**A**) Colon cancer cell lines were subjected to immunofluorescence analysis with indicated antibodies, and CD133- and CD44-double-positive populations in each immunstained tumor cell were quantified. Nuclei were counter-stained with DAPI. Images were captured using a Zeiss confocal microscope. Scale bars: 25 μm. (**B**) D-*K-Ras* MT cells harboring mutation both *APC* and *PI3K* were subcutaneously injected into flanks of BALB/c nude mice (*n* = 4). Mice were treated for 21 days by intra peritoneal (I.P.) injection with vehicle or with 25 mg/kg CDP0857 once every three days. Gross images showing representative tumor cases treated with drugs, with enlarged tumor images for each case presented at the bottom. (**C**) Tumor volumes of mice treated with vehicle or CDP0857 were measured with the Vernier calipers every 3 days. (**D**) Paraformaldehyde (PFA)-fixed paraffin sections from excised tumors were incubated with antibodies against β-catenin, pan-Ras, pERK, PCNA, CD133, CD44, and EpCAM. Images were captured using a Zeiss confocal microscope. Representative images were selected from at least 3 different fields. Scale bars, 50 μm. (**E**) Expression of antibodies in the immunostained tumor tissues in each representative xenograft tumor was quantified using ImageJ software. All data are shown as mean ± SD for at least 3 independent specimens. ** *p*  < 0.005, *** *p*  < 0.0005, 2-sided *t* test.

## Data Availability

All data pertaining to the study described in the manuscript are described in the report.

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
