# Peer review of "A Small Molecule Promoting Neural Differentiation Suppresses Cancer Stem Cells in Colorectal Cancer"

_biomedicines, 2022, doi:10.3390/biomedicines10040859_

Round 1
Reviewer 1 Report
In this manuscript, the authors showed that CPD0857 reduces neural progenitor cell proliferation while increasing neuronal differentiation. CDP0857 effectively reduced CSC marker expression and suppressed self-renewal capacity. CPD0857-treated xenograft mice showed tumor growth regression and a decrease in the number of CSCs in tumors. This is an interesting study. The following points should be addressed to improve the manuscript,
- The novelty of the study should be clearly emphasized.
- Include densitometry in Fig. 1C.
- Use arrows to clearly indicate the specific bands of interest in Fig 1G.
- In Fig 2D, the number of cells should be almost equal among groups in order to compare the expression levels of nestin.
- In Fig 3A, the magnification of cells among groups are different.
- The localization of beta-catenin should be analyzed in Fig. 4E
Author Response
COMMENTS FOR THE AUTHOR:
biomedicines-1645710
Reviewer #1: In this manuscript, the authors showed that CPD0857 reduces neural progenitor cell proliferation while increasing neuronal differentiation. CDP0857 effectively reduced CSC marker expression and suppressed self-renewal capacity. CPD0857-treated xenograft mice showed tumor growth regression and a decrease in the number of CSCs in tumors. This is an interesting study. The following points should be addressed to improve the manuscript:
1. The novelty of the study should be clearly emphasized.
We agree with the reviewer’s comments. Our manuscript has the following four key points. 1) CPD0857 promotes neuronal differentiation of neural progenitor cells (NPCs) that have the multipotent capacity and self-renewal activity which is remarkably similar to cancer stem cell (CSC), 2) CPD0857 inhibits both Wnt/β-catenin and Ras-ERK signaling by ubiquitin-dependent proteasomal degradation of β-catenin and Ras protein in NPCs, 3) CDP0857 reduces expression of CSC markers and suppresses self-renewal capacity in CRCs, 4) CPD0857 inhibits proliferation and expression of CSC markers in D-K-Ras MT cells carrying K-Ras, APC and PI3K mutations, indicating inhibition of Wnt/β-catenin, Ras-ERK, and PI3K/AKT signaling pathways. As reviewers noted, we have addressed these significances in our revised manuscript.
2) Include densitometry in Fig. 1C.
We thank the reviewer for this comment. Reviewer can find the densitometry result of Fig. 1C in Fig. 1D of the original manuscript.
3) Use arrows to clearly indicate the specific bands of interest in Fig 1G.
We apologize for not being clear about this. As suggested, we have added an indication to designate specific bands (Fig. 1G in the original figure, Fig. 1H in the revised figure). Moreover, we have also added molecular weight information of the indicated western blot bands in Fig. 1C, 1F, and 1H of our revised manuscript.
4) In Fig 2D, the number of cells should be almost equal among groups in order to compare the expression levels of nestin.
We agree with the reviewer’s comments. However, as seen in Fig. 2A, when CDP0857 was treated for one day, the number of NPC was reduced by inhibiting proliferation and promoting differentiation of NPCs. Moreover, the number of NPCs was significantly reduced when CPD0857 was treated for two days as seen in Fig. 2D. Therefore, it is not surprising to show a reduced number of NPCs in the CPD0857 treated group.
5) In Fig 3A, the magnification of cells among groups is different.
We apologize for the lack of clarity. We used images with the same magnificence in Fig. 3A. To avoid confusion, we have added a scale bar in all bright and fluorescence images in Fig. 1A, 2A, 2D, 2G, 3A, 3C, 4A, and 4D of our revised manuscript.
6) The localization of beta-catenin should be analyzed in Fig. 4E
The nuclear localization of β-catenin and subsequent inappropriate activation of TCF/LEF-mediated transcription may be an important process in both the establishment and maintenance of cancer stem cells [1]. Here, therefore, we have compared the expression of nuclear-localized β-catenin in vehicle or CPD0857 treated groups in Fig. 4E. This is an important issue in this field, therefore, we have added the above reference in the revised manuscript.
References
- Morgan, R.G., Ridsdale, J., Tonks, A., and Darley, R.L. (2014). Factors affecting the nuclear localization of beta-catenin in normal and malignant tissue. Journal of cellular biochemistry 115, 1351-1361.
Reviewer 2 Report
biomedicines-1645710
In this paper, the authors studied a small molecule, CPD0857, targeting canonical WNT and Ras/ERK and its effects on NPCs and CRC cells. This is an interesting study with clinical potential. From the perspective of academic criticism, several technical concerns need to be addressed to further improve the quality of this manuscript, as appended below.
All the abbreviations need to be clarified at their first appearance, for example, ISCs was not clarified. Some of the terms were not correctly used, for example, “nerve cells” should be corrected to “neural cells” or “neuron”. Please do a throughout grammar and spell check and correct all the errors in the manuscript (use some professional help if necessary).
The citation for “…we performed a high throughput screen (HTS) using a dual cell-based screening system to identify a small molecule...” is missing. This should be the most important background for the current work, please add the reference and provide a brief summary on the “dual cell-based screening system”.
Scale bars should be added to all the phase-contrast and fluorescence images.
For all the western-blot data, the molecular weight of the indicated band should be labeled, and the correlated quantitative data should be supplemented.
In Figure 1F, the band of beta-catenin on 10, 25uM CPD0857 are not uniform. Figure 1G is low-quality and not readable. There are apparent issue of membrane-transformation and blocking during the western-blot. This dataset validates the fundamental function of drug. Please replace the figure.
Figure 2A is not readable. Please re-adjust the size of the figure.
Figure 4E should be replaced by higher-amplification images.
In the animal study, the flow data showed a <10% drop on the CSC population, but the inhibition on tumor growth seemed to be much more effective with a > 50% decrease on the tumor burden. How does the author explain the result? A drop on Epcam suggested a potential EMT in tumor. Was the Ki67 expression on CD44+CD133+ cells measured? What did the survival curves look like?
The two parts of the story, NPC differentiation and CRC treatment, seem to be very independent to each other. Why did the author put them together but not develop them into two papers? In fact, the paper looks like a stack of data rather than a logical story. How can the authors improve this?
Author Response
COMMENTS FOR THE AUTHOR:
biomedicines-1645710
Reviewer #2: In this paper, the authors studied a small molecule, CPD0857, targeting canonical WNT and Ras/ERK and its effects on NPCs and CRC cells. This is an interesting study with clinical potential. From the perspective of academic criticism, several technical concerns need to be addressed to further improve the quality of this manuscript, as appended below.
1) All the abbreviations need to be clarified at their first appearance, for example, ISCs were not clarified.
We apologize for the lack of clarity. ISC is an intestinal stem cell and the reviewer can find it in the second line of the Introduction of the original manuscript. Moreover, we have also clarified all abbreviations seen in the result and figure legend sections of the revised manuscript.
2) Some of the terms were not correctly used, for example, “nerve cells” should be corrected to “neural cells” or “neuron”.
We apologize for this error. As noted, we have now carefully checked the manuscript and have corrected all mistakes in the revised manuscript.
3) Please do a throughout grammar and spell check and correct all the errors in the manuscript (use some professional help if necessary).
As noted, we have corrected the sentence. Moreover, we have also carefully checked the entire manuscript to correct the English grammar and style by Dr. Elise Lamar who is a professional medical editor.
4) The citation for “…we performed a high throughput screen (HTS) using a dual cell-based screening system to identify a small molecule...” is missing. This should be the most important background for the current work, please add the reference and provide a brief summary on the “dual cell-based screening system”.
As suggested, we have added a reference deciphering HTS using a dual cell-based screening system in the revised manuscript.
5) Scale bars should be added to all the phase-contrast and fluorescence images.
A similar comment was raised by reviewer 1. We have addressed it in the response to comment 5 from reviewer 1. Briefly, we have added a scale bar in all phase-contrast and fluorescence images in Fig. 1A, 2A, 2D, 2G, 3A, 3C, 4A, and 4D of our revised manuscript.
6) For all the western-blot data, the molecular weight of the indicated band should be labeled, and the correlated quantitative data should be supplemented.
A similar comment was raised by reviewer 1. We have addressed it in the response to comment 3 from reviewer 1. Briefly, we have added molecular weight information of the indicated western blot bands in Fig. 1C, 1F, and 1H of our revised manuscript. Moreover, we also have provided quantitative data of western blot results in Fig. 1D, 1G, 1I, and 1J of the revised manuscript.
7) In Figure 1F, the band of beta-catenin on 10, 25uM CPD0857 are not uniform. Figure 1G is low-quality and not readable. There are apparent issue of membrane-transformation and blocking during the western-blot. This dataset validates the fundamental function of drug. Please replace the figure.
We agree with the reviewer’s comments. As the reviewer noted, we performed this experiment again and ran the gel contemporaneously. The new figures were presented in Fig. 1F and 1H of the revised manuscript.
8) Figure 2A is not readable. Please re-adjust the size of the figure.
We thank the reviewer for kindly pointing this out. As requested, we have re-adjusted the size of the original images and provided this in Fig. 2A and supplementary Fig. 1 of the revised manuscript.
9) Figure 4E should be replaced by higher-amplification images.
As requested, we have corrected the immunostaining pictures and provided higher-amplification images in Fig. 4D of the revised manuscript.
10) In the animal study, the flow data showed a <10% drop on the CSC population, but the inhibition on tumor growth seemed to be much more effective with a > 50% decrease on the tumor burden. How does the author explain the result? A drop on Epcam suggested a potential EMT in tumor. Was the Ki67 expression on CD44+CD133+ cells measured? What did the survival curves look like?
We thank the reviewer for kindly pointing this out and also apologize for not being clear about this. CPD0857 inhibits CRC progression through various mechanisms such as reducing CSC population and cell proliferation, and promoting cell apoptosis seen in our manuscript and previously published study [1]. Therefore, it is not surprising that cancer inhibition of more than 50% was observed despite a reduction of only 10% of CSCs. However, to avoid confusion, we have deleted Fig. 4D in the original figure. In a recent paper, EpCAM was also used as a reliable marker of colorectal CSCs [2]. Here, therefore, we used this as a marker of colorectal CSCs. In the case of ki67 expression on CD44&CD133 double-positive cell and drawing survival curve, we definitely agree with these points and as a further study, we will perform these. Instead, to confirm the effect of CPD0857 on the inhibition of CSC proliferation in vitro, we have quantified CD133 and CD44 positive cell numbers after CPD0857 treatment on D-K-Ras MT CRCs which have a highly accumulated CSC population (Fig. 3C-E). Moreover, to investigate its role in vivo, we did quantify PCNA-positive cells, another cell proliferation marker, in xenograft tumor samples. The new modified figures were presented in Fig. 4D and 4E of the revised manuscript.
11) The two parts of the story, NPC differentiation and CRC treatment, seem to be very independent to each other. Why did the author put them together but not develop them into two papers? In fact, the paper looks like a stack of data rather than a logical story. How can the authors improve this?
The reviewer is right. As the reviewer noted, regulation of NPC differentiation and CRC treatment seem to be independent, however, NSCs have a multipotent capacity and self-renewal activity that is remarkably similar to cancer stem cells [3]. Especially, CD133 cell surface glycoprotein used as a cancer stem cell marker for CRCs is also used for identifying neural stem cells [4]. In our previous study, CPD0857 inhibited both Wnt/β-catenin and Ras-ERK signaling by ubiquitin-dependent proteasomal degradation of β-catenin and Ras protein in colorectal cancer cell lines [1]. Here, we also observed that CPD0857 potentially could inhibit the growth of colorectal cancer stem cells through inhibition of both Wnt/β-catenin and Ras/ERK pathways. Moreover, we discovered that CDP0857 reduces stemness characteristics of both CRC CSCs and NPCs by reducing self-renewal capacities of CSCs and promoting neuronal differentiation of NPCs, respectively. Furthermore, recent findings suggested that colorectal cancer stem cells show neuronal differentiation potential and the nerve cell can be derived from carcinoma stem cells and these can contribute to cancer progression [3,5]. Therefore, the regulation of NPCs' self-renewal and differentiation can be the key link to targeting CRC and its CSCs; and support our concept of research. To improve the quality of this, we have added more references in the discussion section of the revised manuscript.
References
- Choi, J.K., Cho, H., and Moon, B.S. (2020). Small Molecule Destabilizer of beta-Catenin and Ras Proteins Antagonizes Growth of K-Ras Mutation-Driven Colorectal Cancers Resistant to EGFR Inhibitors. Targeted oncology 15, 645-657.
- Leng, Z., Xia, Q., Chen, J., Li, Y., Xu, J., Zhao, E., Zheng, H., Ai, W., and Dong, J. (2018). Lgr5+CD44+EpCAM+ Strictly Defines Cancer Stem Cells in Human Colorectal Cancer. Cellular physiology and biochemistry : international journal of experimental cellular physiology, biochemistry, and pharmacology 46, 860-872.
- Xu, L., Zhang, M., Shi, L., Yang, X., Chen, L., Cao, N., Lei, A., and Cao, Y. (2021). Neural stemness contributes to cell tumorigenicity. Cell & bioscience 11, 21.
- Wang, C., Xie, J., Guo, J., Manning, H.C., Gore, J.C., and Guo, N. (2012). Evaluation of CD44 and CD133 as cancer stem cell markers for colorectal cancer. Oncology reports 28, 1301-1308.
- Lu, R., Fan, C., Shangguan, W., Liu, Y., Li, Y., Shang, Y., Yin, D., Zhang, S., Huang, Q., Li, X., et al. (2017). Neurons generated from carcinoma stem cells support cancer progression. Signal transduction and targeted therapy 2, 16036.
Round 2
Reviewer 1 Report
The authors have addressed all the comments and the manuscript can be accepted for publication.
Reviewer 2 Report
Accept in present form